# Concurrent optoacoustic tomography and magnetic resonance imaging of resting-state functional connectivity in the mouse brain

Irmak Gezginer [1,2,6], Zhenyue Chen[1,2,3,6], Hikari A. I. Yoshihara [1,2], Xosé Luís Deán-Ben[1,2], Valerio Zerbi[4,5] & Daniel Razansky [1,2] ✉

Resting-state functional connectivity (rsFC) has been essential to elucidate the intricacy of brain organization, further revealing clinical biomarkers of neurological disorders. Although functional magnetic resonance imaging (fMRI) remains a cornerstone in the field of rsFC recordings, its interpretation is often hindered by the convoluted physiological origin of the blood-oxygen-level-dependent (BOLD) contrast affected by multiple factors. Here, we capitalize on the unique concurrent multiparametric hemodynamic recordings of a hybrid magnetic resonance optoacoustic tomography platform to comprehensively characterize rsFC in female mice. The unique blood oxygenation readings and high spatio-temporal resolution at depths provided by functional optoacoustic (fOA) imaging offer an effective means for elucidating the connection between BOLD and hemoglobin responses. Seed-based and independent component analyses reveal spatially overlapping bilateral correlations between the fMRI-BOLD readings and the multiple hemodynamic components measured with fOA but also subtle discrepancies, particularly in anti-correlations. Notably, total hemoglobin and oxygenated hemoglobin components are found to exhibit stronger correlation with BOLD than deoxygenated hemoglobin, challenging conventional assumptions on the BOLD signal origin.

Analysis of resting-state functional connectivity (rsFC) has been a transformative tool in neuroscience and clinical investigations, providing new insights into brain organization and significantly advancing our comprehension of cerebral dynamics[1]. The method provides a window into the functional architecture of the brain through the analysis of correlation patterns in spontaneous fluctuations associated with neural activity[2]. Studies of rsFC have contributed to the identification of brain networks and revealed how these develop over the lifespan[3], while also providing the basis for a better understanding of individual differences in behavior and cognition[4,5]. rsFC has also played a critical role in elucidating the pathophysiology of various psychiatric and neurological conditions, such as Alzheimer's disease[6–8], Parkinson's disease[9,10], and epilepsy[11,12], and it has been used to study the effects of brain injury and recovery mechanisms[13,14]. Alterations in the patterns of spatial coherence associated with disrupted connectivity further facilitate the identification of biomarkers that can improve diagnosis and treatment monitoring of neurological and psychiatric conditions[14–16].

[1]Institute for Biomedical Engineering and Institute of Pharmacology and Toxicology, Faculty of Medicine, University of Zurich, Zurich, Switzerland. [2]Institute for Biomedical Engineering, Department of Information Technology and Electrical Engineering, ETH Zurich, Zurich, Switzerland. [3]Institute of Precision Optical Engineering, School of Physics Science and Engineering, Tongji University, Shanghai, China. [4]Department of Psychiatry, Faculty of Medicine, University of Geneva, Geneva, Switzerland. [5]Department of Basic Neurosciences, Faculty of Medicine, University of Geneva, Geneva, Switzerland. [6]These authors contributed equally: Irmak Gezginer, Zhenyue Chen. ✉e-mail: daniel.razansky@uzh.ch

By measuring the spontaneous low-frequency fluctuations in the blood-oxygen-level-dependent (BOLD) signal, functional magnetic resonance imaging (fMRI) has been the mainstay in the realm of rsFC studies owing to its unparalleled ability to map and analyze the intricate neural networks of the entire brain noninvasively[17,18]. Yet, accurate interpretation of these signals remains elusive as they depend on multiple factors such as blood flow, blood volume, metabolic rate of oxygen consumption, or the baseline physiological state[19]. Over the past decades, other non-invasive or minimally invasive methods have further been used to shed new light on rsFC patterns[20]. For instance, electro-encephalography (EEG) measures the electrical activity of the brain at rest with remarkable temporal resolution in the millisecond range, but its application is often impeded by poor spatial resolution[21,22]. Resting-state PET is likewise constrained by its inherently limited spatial and temporal resolution, particularly in preclinical studies[23,24], whilst its reliance on ionizing radiation further restricts repeated use. Functional ultrasound (fUS) has recently emerged as a valuable tool for rsFC studies, offering high spatial and temporal resolution to map cerebral blood flow dynamics associated with neuronal activity[25,26]. However, its main blood flow-related contrast, as well as strong acoustic attenuation and aberrations in the skull limit the fUS applicability in neuroscience[27,28]. Macroscopic optical imaging techniques, including intrinsic signal optical imaging (ISOI)[29], near-infrared spectroscopy (NIRS)[30,31], and epi-fluorescence calcium imaging[32,33], offer diverse brain activity readings and molecular-specific contrast but the strong light scattering in biological tissues restricts the investigations to superficial brain areas or otherwise results in strong spatial resolution degradation with depth. Optoacoustic (OA) imaging synergistically combines stimulation with light and ultrasound detection to offer a unique perspective on brain function[34,35]. By capitalizing on the distinct absorption spectra of oxygenated (HbO) and deoxygenated (HbR) forms of hemoglobin, functional optoacoustic (fOA) imaging provides otherwise unattainable multiparametric hemodynamic readings with excellent spatiotemporal resolution[36], further offering high sensitivity detection of stimulus-evoked activity and rsFC in murine models[37–40], and humans[41]. As with fUS, fOA is affected by the skull acoustic distortions, albeit to a lesser extent as it only involves unidirectional propagation of ultrasound.

To this end, multimodal functional brain imaging has served to corroborate the existence of connectivity patterns and has further contributed to a better understanding of resting-state fMRI (rs-fMRI) measurements. Traditionally, BOLD signals have primarily been associated with paramagnetic HbR levels[42]. However, differential measurements of HbO and HbR responses performed with NIRS or fOA have revealed weaker associations between BOLD and HbR during task- and sensory-related brain activation[36,43]. Here we capitalize on the unique concurrent multiparametric hemodynamic readings of hybrid magnetic resonance optoacoustic tomography (MROT)[36–38] to comprehensively characterize rsFC in mice. MROT delivers simultaneous fMRI and fOA readings from the whole mouse brain noninvasively, thus offering a versatile platform for studying brain function and cross-modality validation. By capturing multiple hemodynamic readings in real time with comparable spatial resolution and volumetric coverage, we provide a holistic approach to unraveling the complexities of neurovascular coupling. We then delve into resting-state fOA (rs-fOA) using seed-based and independent component analysis (ICA) analysis of multiple hemodynamic components and further investigate the intricate relationship between the BOLD-derived rsFC patterns and those obtained by an independent analysis of the individual HbO, HbR, and total hemoglobin (HbT) components.

## Results

### Concurrent fOA and fMRI discern consistent rsFC patterns in the murine brain

The hybrid MROT system features a fOA module inserted into the bore of a 9.4 T preclinical MRI scanner to enable concurrent detection of multispectral OA and BOLD data (see Methods and Supplementary Fig. 1). For each mouse ($n = 16$), $T_1$-weighted, magnetic resonance angiography (MRA), temporally registered rs-fMRI, and rs-fOA datasets were acquired (Fig. 1a). The multimodal datasets were acquired simultaneously and all synchronized functional scans had identical durations across the different modalities, ensuring uniform temporal alignment. The fOA time series were reconstructed using a model-based algorithm incorporating a non-negative constraint in the iterative inversion procedure[44,45] to suppress negative values and other artefacts commonly present in the images reconstructed with standard filtered backprojection algorithms[46]. Time-resolved distributions of HbO, HbR, and HbT hemoglobin concentrations (Fig. 1b) were estimated leveraging the spectroscopic fOA data recorded at five individual wavelengths (700, 730, 755, 800, 850 nm, see "Methods"). Coregistration of MRI and OA data was done by exploiting the common vascular contrast in the OA and MRA images[38]. Multimodal data were normalized to the Allen Mouse Brain Common Coordinate Framework (CCFv3)[47] (Fig. 1c), thus facilitating the analysis of rsFC with each modality and their direct comparison (Fig. 1d).

The rsFC patterns of BOLD, HbO, HbR, and HbT were first calculated using seed-based methods. Seeds representing various cortical regions were positioned on each hemisphere, including the somatosensory, motor, visual, parietal, and retrosplenial areas. The dataset quality was evaluated by examining the concurrent presence of strong inter-hemispheric connectivity in the sensory cortices along with either weak connectivity or anti-correlation between the sensory and the anterior cingulate areas[48,49]. Consequently, of 16 datasets, 10 exhibiting specific FC were included in the analysis (Supplementary Fig. 2). Averaged time series data (Supplementary Fig. 3) were extracted from each pre-processed seed volume to calculate the Pearson's correlation coefficient with other voxels. This served to construct seed-based maps for each hemodynamic component. Strong correlations were observed both intra- and inter-hemispherically across all hemodynamic components at the group level (Fig. 2a). These correlations showed a high degree of spatial overlap between the BOLD and multiparametric OA maps. This spatial overlap was consistently maintained across time (Supplementary Fig. 4). Moreover, robust correlations were observed between the seed maps generated at different time points within each hemodynamic component, indicating consistent rsFC patterns between the first and second part of the time-resolved data (Supplementary Fig. 4). The correlation between the first and second halves was notably lower for HbR and BOLD compared to HbO and HbT (Supplementary Fig. 4), suggesting that HbR, like BOLD, may be more susceptible to noise. The seed maps remained consistent regardless of the presence or absence of magnetic gradients (Supplementary Fig. 5). The use of filtered backprojection image reconstruction resulted in non-specific, non-local rsFC across all components (Supplementary Fig. 6), highlighting the importance of the quantitative non-negative-constrained model-based inversion. We also observed variations in the distribution of correlations within the brain ($z > 2$, CCFv3), which differed among functional regions and hemodynamic components (Fig. 2b). We quantified the spatial smoothness and noise levels of the fOA seed-maps by computing local variance[50] and gradient-based spatial consistency. Consistent with previous findings[29], HbR seed-maps demonstrated significantly higher local variance (mean = $(4.50 \pm 0.09) \times 10^{-3}$) than HbO (mean = $(3.20 \pm 0.14) \times 10^{-3}$, paired $t$ test, t(139) = 9.984, $p = 4.981 \times 10^{-18}$, Cohen's $d = 0.792$, 95% CI = [0.549, 1.035]) and HbT (mean = $(3.60 \pm 0.17) \times 10^{-3}$, paired $t$ test, t(139) = 4.746, $p = 5.091 \times 10^{-6}$, Cohen's $d = 0.510$, 95% CI = [0.272, 0.748]), suggesting noisier connectivity patterns with lower spatial coherence (Supplementary Fig. 7). HbR also showed higher mean spatial gradient (mean = $0.038 \pm 0.0008$), indicating larger fluctuations than HbO (mean = $0.033 \pm 0.0006$, paired $t$ test, t(139) = 12.277, $p = 6.320 \times 10^{-24}$, Cohen's $d = 0.611$, 95% CI = [0.372, 0.850]) and HbT (mean = $0.034 \pm 0.0006$,

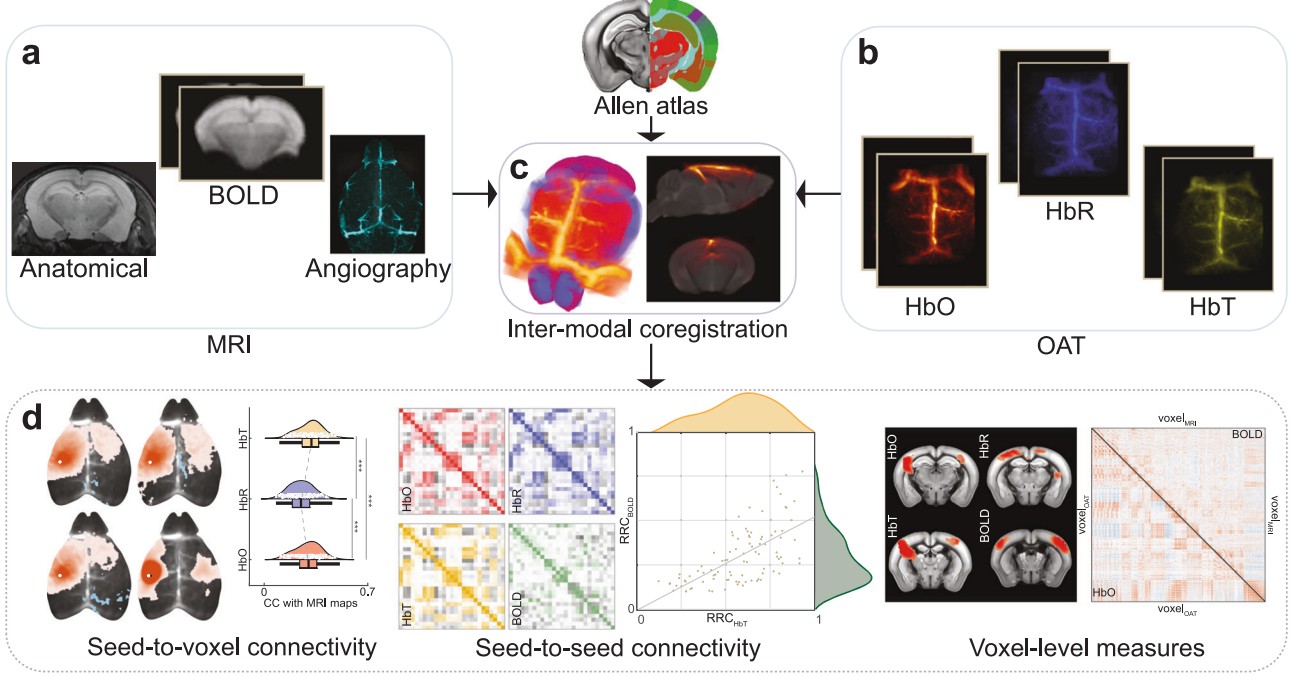

**Fig. 1 | Simultaneous resting-state functional connectivity (rsFC) mapping with functional optoacoustic imaging (fOA) and functional magnetic resonance imaging (fMRI) using the hybrid magnetic resonance optoacoustic tomography (MROT) platform. a** $T_1$-weighted image, magnetic resonance angiography (MRA), and blood oxygen level-dependent (BOLD) signal time series. **b** Time-resolved oxygenated (HbO), deoxygenated (HbR), and total (HbT) hemoglobin concentrations rendered by the multispectral fOA imaging. **c** Coregistration and spatial normalization techniques further facilitate the parcellation of the multi-modal data, delineating distinct anatomical and functional brain regions for comprehensive analysis. **d** The method enables investigations into the correlation of rsFC patterns derived by BOLD versus the different hemoglobin-related optoacoustic components through seed-based and voxel-based analysis methods.

paired $t$ test, t(139) = 5.554, $p$ = 1.371 x $10^{-7}$, Cohen's $d$ = 0.409, 95% CI = [0.173, 0.645]) (Supplementary Fig. 7).

## fOA discerns the rsFC links between distinct hemoglobin components and BOLD

The concurrent acquisition capability of MROT allows for a direct comparison between rsFC measurements obtained from BOLD and the distinct fOA components. BOLD signals have traditionally been associated with a reduced distortion of the local magnetic field corresponding to a decrease in HbR. Yet, a more comprehensive characterization of hemoglobin changes can shed light on the basic mechanisms underlying BOLD responses. Accordingly, we investigated the voxel-wise correlation of BOLD and fOA seed-maps. The results exhibited a strong linear correlation between both modalities. Specifically, BOLD (for $r_{FC}$ > 0.1) demonstrated a more robust correlation with HbT, yielding linear correlation coefficients of 0.71, 0.62, and 0.81 for primary somatosensory, primary motor, and primary visual area seeds, respectively (Fig. 2c–e). In contrast, the weakest correlation with BOLD was observed for HbR, with correlation coefficients of 0.66, 0.59, and 0.76 for the corresponding seeds. This trend of linear dependency between BOLD and fOA was consistent for the remaining seeds from the left hemisphere, including primary somatosensory- lower limb, secondary motor, parietal, and retrosplenial areas (Supplementary Fig. 8). Overall, HbT and HbO featured stronger correlations with BOLD compared to HbR across all seeds, with HbT exhibiting the highest correlation in six seed-maps and HbO leading in one. When examining these correlations across the first and second halves of the data, we observed that the voxel-wise correlations between the fOA components and BOLD were lower when considering only half of the acquisition duration, potentially due to increased noise and variability in shorter time windows (Supplementary Fig. 9). To comprehensively evaluate the correlations between BOLD and the hemodynamic components HbO, HbR, and HbT, we further analyzed linear correlation coefficient (r) distributions derived from the defined seeds considering the whole range of FC ($|r_{FC}|$ <1) at the group level. The results suggested that both HbT (mean = 0.314, median = 0.331) and HbO (mean = 0.318, median = 0.325) exhibit a more robust correlation with BOLD compared to HbR (mean = 0.308, median = 0.297) (Fig. 2f).

We extended our analysis to investigate the spatial correlation between BOLD and fOA seed maps by calculating the normalized cross-correlation (CC) of these maps across animals. This examination aimed to discern individual-level differences in positive and negative connections and spanned two sub-categories: seed maps containing positive correlations, and those including only anti-correlations. Our findings at the group level revealed a minor effect of stronger correlation in positive connections between the BOLD seed maps and both HbT (mean = 0.586 ± 0.006, paired $t$ test, t(127) = 3.691, $p$ = 3.298 × $10^{-4}$, Cohen's $d$ = 0.161, 95% CI = [0.070, 0.252]) and HbO (mean = 0.585 ± 0.006, paired $t$ test, t(127) = 4.576, $p$ = 1.115 × $10^{-5}$, Cohen's $d$ = 0.144, 95% CI = [0.080, 0.208]) as compared to HbR (mean = 0.575 ± 0.006) (Fig. 2g and Supplementary Fig. 10). Interestingly, in negative connections, we observed a substantial effect with HbT (mean = 0.302 ± 0.007, paired $t$ test, t(127) = 11.578, $p$ = 1.323 × $10^{-21}$, Cohen's $d$ = 0.577, 95% CI = [0.449, 0.705]) and HbO (mean = 0.288 ± 0.008, paired $t$ test, t(127) = 9.631, $p$ = 8.129 × $10^{-17}$, Cohen's $d$ = 0.390, 95% CI = [0.300, 0.480]), indicating a higher level of similarity than for HbR (mean = 0.254 ± 0.007) (Fig. 2h and Supplementary Fig. 10). In addition, HbT exhibited a superior correlation with the BOLD seed maps compared to HbO for negative connections (paired $t$ test, t(127) = 5.689, $p$ = 8.384 × $10^{-8}$, Cohen's $d$ = 0.165, 95% CI = [0.105, 0.225]) (Fig. 2h and Supplementary Fig. 10). These effects were more pronounced in regions with less attenuation, specifically at depths closer to the brain surface (Supplementary Fig. 11). Significantly, higher CC in positive connections was evident for HbT

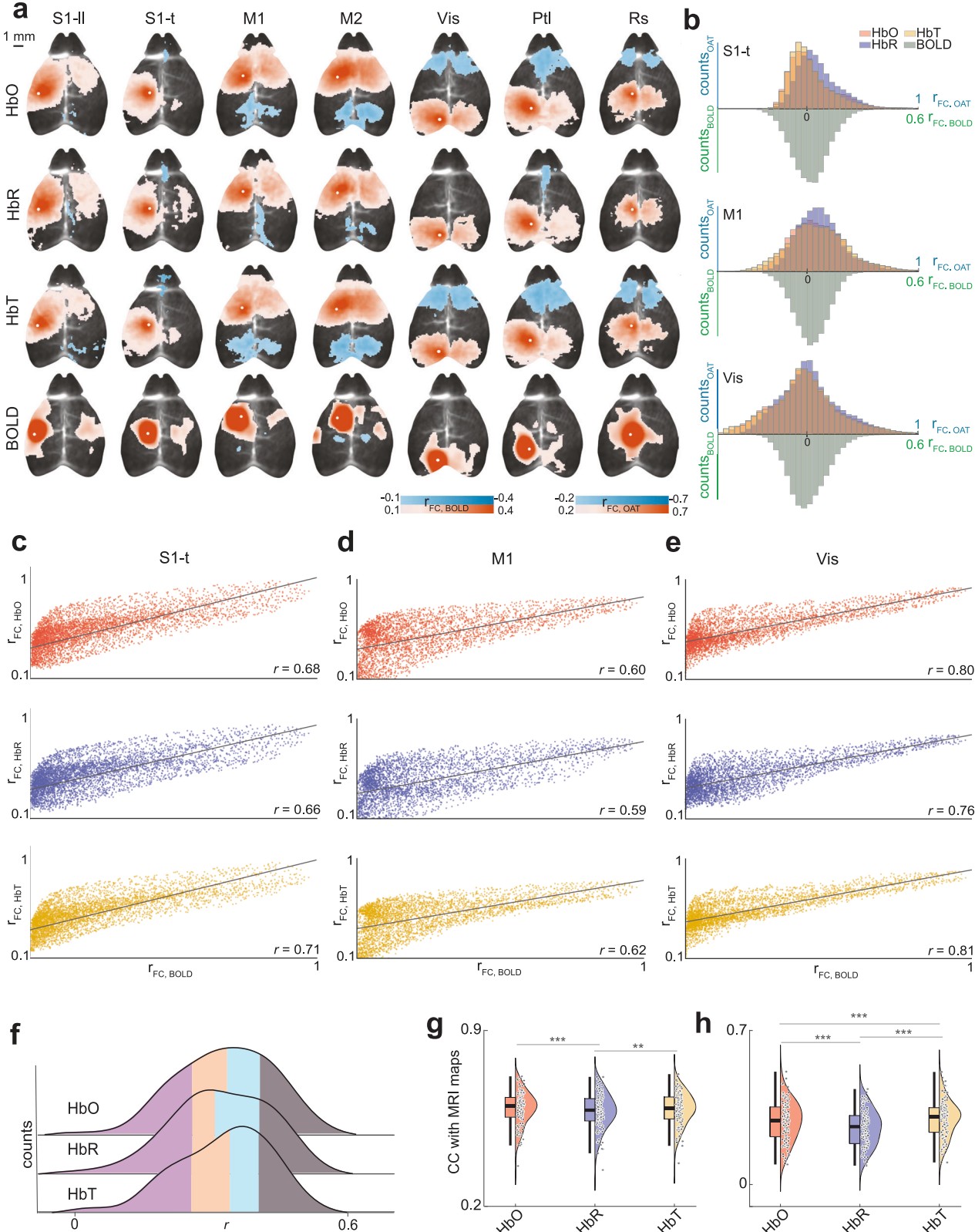

and HbO (paired *t* test, *p* < 0.05) than for HbR, as observed in four and three out of 10 individual subjects, respectively Supplementary Figs. 12, 14). Conversely, for negative connections, all animals showed a tighter correlation with HbT compared to HbR. Similarly, stronger correlations to HbO in comparison to HbR were observed in all but two animals. Moreover, five animals exhibited a higher CC to HbT than HbO (Supplementary Figs. 13, 14). Importantly, this trend of stronger correlations

for HbT and HbO compared to HbR was consistently observed over time (Supplementary Fig. 15) and remained robust across different fluence correction strategies (Supplementary Figs. 16–18).

**BOLD validates fOA connectivity across functional brain regions**
rsFC analysis was then extended beyond voxel-level correlations to explore region-to-region connectivity patterns, thus providing insights

**Fig. 2 | Multiparametric functional connectivity and group-level correlations (n = 10) between the simultaneously acquired blood oxygen level-dependent (BOLD), oxygenated (HbO), deoxygenated (HbR), and total (HbT) hemoglobin components. a** Group seed-maps derived from BOLD, HbO, HbR, and HbT for seven functional regions positioned in the left hemisphere overlaid onto the structural optoacoustic image that emphasizes major vascular structures. Strong intra- and inter-hemispheric correlations were evident, revealing spatial overlap and distinct anti-correlations. White circles represent the seed regions. S1-t, primary somatosensory area, trunk; S1-ll, primary somatosensory area, lower limb; M1, primary motor cortex; M2, secondary motor cortex; Vis, primary visual area; Ptl, parietal area; Rs, Retrosplenial area. **b** Distribution of correlation coefficients within the brain for three representative seeds across the various hemodynamic components. **c–e** Voxel-wise relationship of functional connectivity (FC) (r > 0.1) derived from group seed maps of three representative seeds. The inter-modal correlation differed across the functional regions and hemoglobin components. The highest

correlation to BOLD was observed for HbT in S1-t, M1, and Vis seeds, followed by HbO and HbR. **f** Distribution of linear correlation coefficients (r) between voxel-wise BOLD and HbO/HbR/HbT for the whole range of FC. The distinct colors represent the quartiles of each distribution. **g** Cross-correlation (CC) analysis of positive connections (n = 128) between BOLD seed maps and HbO, HbR, HbT, indicating higher correlations with HbT ($p = 3.298 \times 10^{-4}$) and HbO ($p = 1.115 \times 10^{-5}$) as compared to HbR. (**h**) Evaluation of negative connections (n = 128) highlighting stronger correlations between BOLD seed maps and HbT ($p = 1.323 \times 10^{-21}$), HbO ($p = 8.129 \times 10^{-17}$), as compared to HbR, with HbT exhibiting superior correlation compared to HbO ($p = 8.384 \times 10^{-8}$) in negative connections. The central line within the box represents the mean, while the lower and upper box edges denote the 25th and 75th percentiles, respectively. The whiskers extend to the furthest data points within the non-outlier range. (*) $p < 0.05$, (**) $p < 0.01$, (***) $p < 0.001$, paired t test (two-sided).

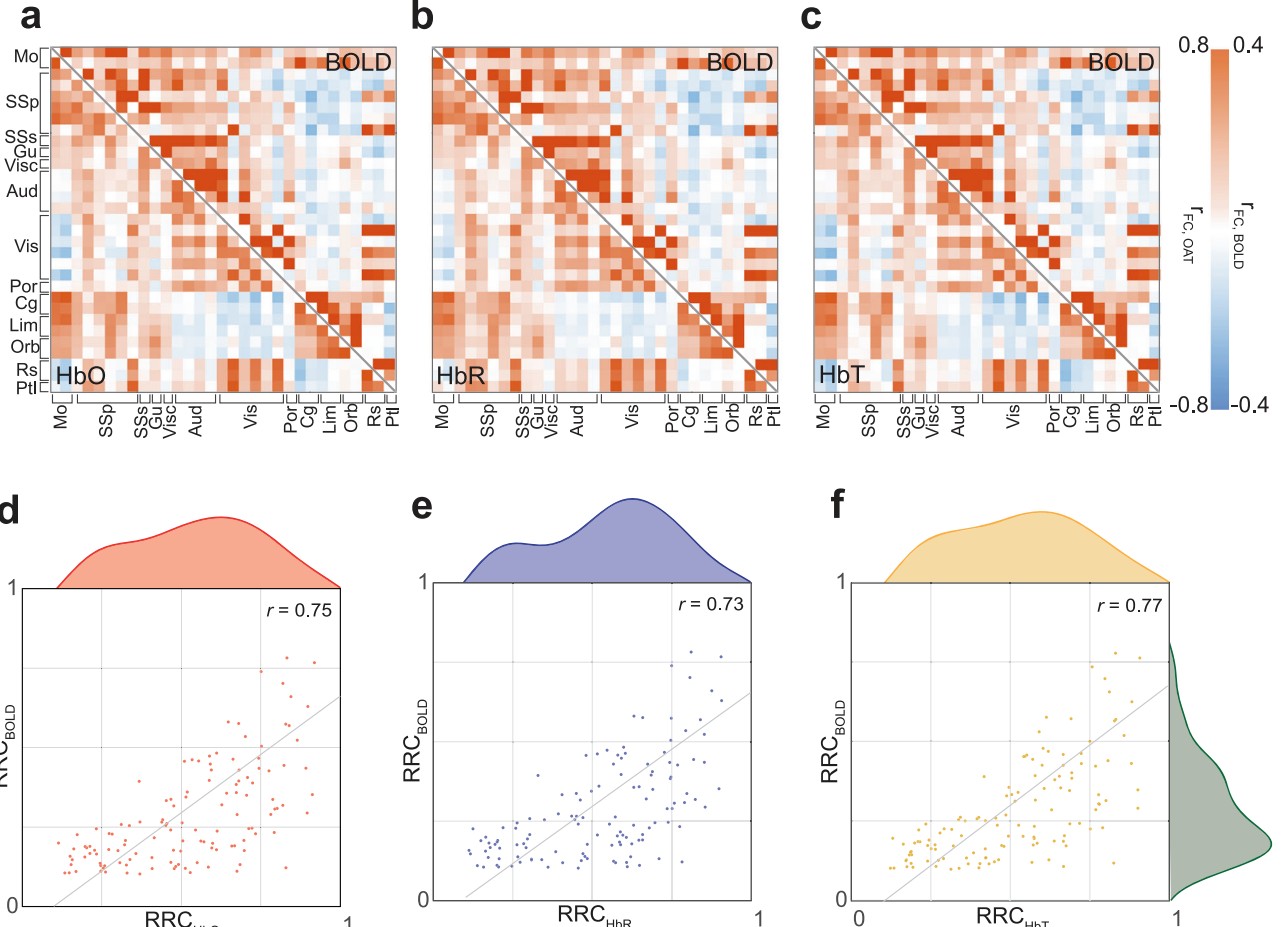

**Fig. 3 | Multimodal region-to-region correlations. a–c** Functional connectivity matrices showcasing correlations between each pair of regions defined based on the Allen common coordinate framework (CCF). The upper triangles show region-to-region correlation (RRC) values derived from the blood oxygen level-dependent (BOLD) data. Lower triangles depict the corresponding RRC values for oxygenated (HbO), deoxygenated (HbR), and total (HbT) hemoglobin. **d–f** Map of the RRC values (r > 0.1) obtained from BOLD versus functional optoacoustic (fOA)-derived

components for connection edges, accompanied by the statistical distribution of the RRC values. Gray solid lines depict the linear fit. Mo = motor areas; SSp = primary somatosensory areas; SSs = supplementary somatosensory areas; Gu = gustatory areas; Visc = visceral areas; Aud = auditory areas; Vis = visual areas; Por = posthinal areas; Cg = anterior cingulate areas; Lim = limbic areas; Orb = orbital areas; Rs = retrosplenial areas; Ptl = parietal areas.

into the network organization of the brain. By examining region-wise correlations, coordinated activity between specific brain areas can be uncovered, shedding light on the functional relationships underlying cognitive processes and behavior. To comprehend the efficacy of fOA in capturing region-wise correlations, we expanded the seed list using the Allen CCF, encompassing various regions such as motor, somatosensory, gustatory, auditory, visual, posthinal, anterior cingulate,

prelimbic, infralimbic, orbital, retrosplenial, and parietal areas (refer to Supplementary Table 1 for the complete list of regions). For each OA-derived hemodynamic component, we generated region-to-region connectivity matrices based on pairwise Pearson's correlations (Fig. 3a–c). Delving deeper into the association between BOLD and fOA signals for HbO, HbR, and HbT, we conducted edge-wise Pearson correlations across region pairs for specific region-wise functional

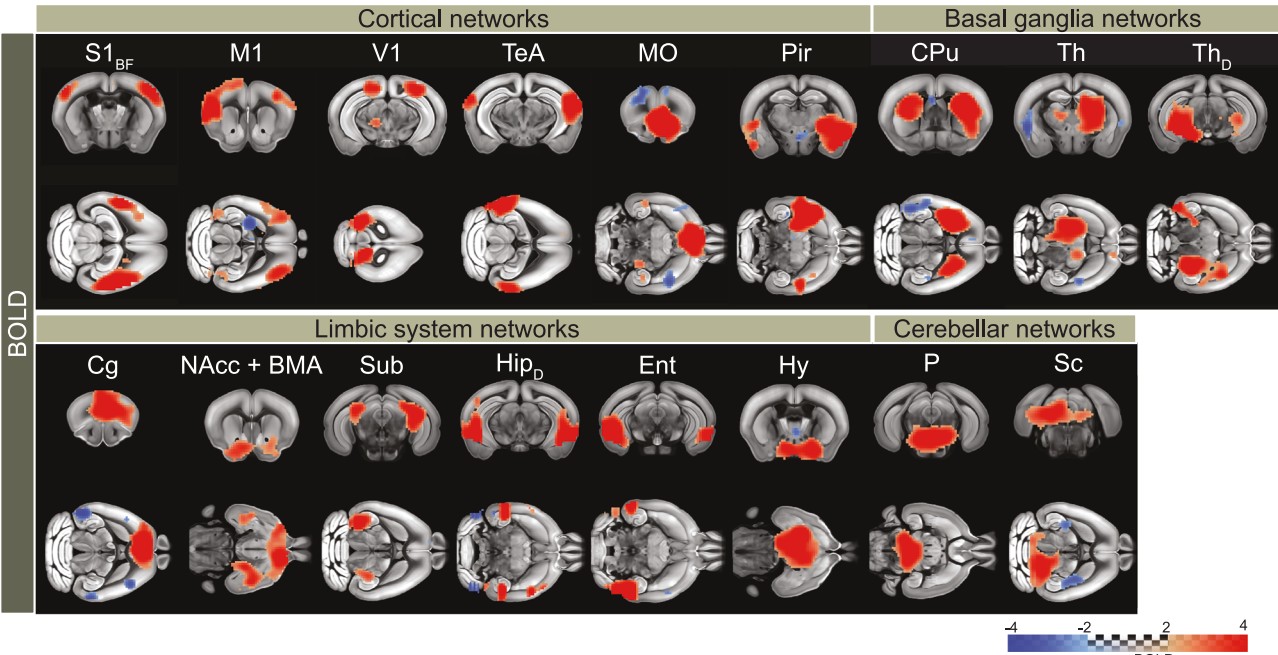

**Fig. 4 | Group independent component analysis (ICA) components classified as networks, which were identified from the blood oxygen level-dependent (BOLD) recordings.** BOLD network patterns were detected in the cortex, basal ganglia, limbic system, and cerebellum. $S1_{BF}$ = primary somatosensory area, barrel field; M1 = primary motor area; V1 = primary visual area; TeA = temporal association area; MO = orbital area, medial part; Pir = piriform area; CPu = caudoputamen; Th = thalamus; $Th_D$ = dorsal thalamus; Cg = anterior cingulate area; NAcc = nucleus accumbens; BMA = basomedial amygdalar nucleus; Sub = Subiculum; $Hip_D$ = dorsal hippocampus; Ent = entorhinal area; Hy = hypothalamus; P = pons; Sc = colliculus.

connectivity ($r_{FC}$ > 0.1). These analyses underscored a robust coupling of HbO ($r$ = 0.75), HbR ($r$ = 0.73), and HbT ($r$ = 0.77) to BOLD while capturing region-to-region connections (Fig. 3d−f). We similarly constructed voxel-by-voxel connectivity matrices (each sized 19000 x 19000) for each OA hemodynamic component, focusing on the gray matter voxels. These matrices uncovered high correlations within distinct functional brain regions, paralleling the BOLD findings (Supplementary Fig. 19).

## Independent component analysis (ICA) identifies cortical and limbic networks rendered by fOA

ICA is a particularly valuable approach for the rsFC analysis as no pre-specified regions of interest or signal profiles are required, thus allowing for the unbiased discovery of resting-state networks. It facilitates the exploration of temporally correlated functional networks by decomposing complex functional data into statistically independent components, thus enabling the identification of distinct patterns of brain activity. Using group-level ICA, we conducted separate analyses for each hemodynamic channel, resulting in the generation of 20 independent components (ICs). Among the BOLD ICs, seventeen exhibited symmetric patterns in meaningful anatomical regions of the mouse brain, denoted as rs-fMRI networks (Fig. 4). These networks spanned across the cortical, basal ganglia, limbic, and cerebellar regions, aligning well with previously reported components[49,51]. The remaining three components were classified as spurious. No components were identified as vascular or motion artifacts for BOLD. Similarly, bilateral networks were discerned from OA hemodynamic components (HbO, HbR, and HbT). 10 out of 20 components were classified as resting-state networks for HbO, HbR, and HbT, encompassing the cortex and the limbic system (Fig. 5). The remaining components were characterized as reconstruction noise and spurious components (Supplementary Fig. 20). Networks such as the primary somatosensory area, secondary motor area, medial orbital area, and retrosplenial area were consistently identified across each of the fOA

components. Notably, some networks were uniquely identified in the HbO (primary somatosensory area, lower limb), HbR (retrosplenial area-ventral part and anterior cingulate area), and HbT (primary visual area and subiculum) components. The commonly identified networks exhibited a high spatial overlap across all the fOA-derived hemodynamic components. In addition, certain networks located in the superior sites of the mouse brain (primary somatosensory area-barrel field, primary visual area, medial orbital area, retrosplenial area, anterior cingulate area) were commonly identified with both BOLD and fOA. Subcortical regions identified by fOA include the hippocampus and subiculum; however, due to penetration depth limitations, subcortical regions present in BOLD-derived ICs were mostly absent from the components identified by fOA.

## Discussion

The intricate interplay of neurovascular coupling and dynamic physiological states has mainly been studied with standalone functional neuroimaging methods, thus only offering a restricted view of complex phenomena. In particular, the interpretation of BOLD signals in fMRI – a mainstay in neuroscience – is confounded by their multifactor dependence[19]. The enhanced information provided by MROT opens new avenues to investigate the basic mechanisms underlying hemodynamic changes associated with neuronal activity. Concurrent acquisition of rs-fMRI and rs-fOA datasets in lightly anesthetized mice allowed for the observation of distinct functional connectivity patterns. Spatially overlapping and temporally stable bilateral correlations of multiple functional regions were observed from HbO, HbR, HbT, and BOLD seed maps, validating fOA as a reliable tool for studying functional connectivity in the murine brain. The successful identification of resting-state networks through ICA of fOA data further underscores and validates its potential for functional connectivity studies. The hybrid MROT imaging approach additionally enabled the investigation of spatiotemporal correlations between the two modalities. Seventeen anatomically meaningful

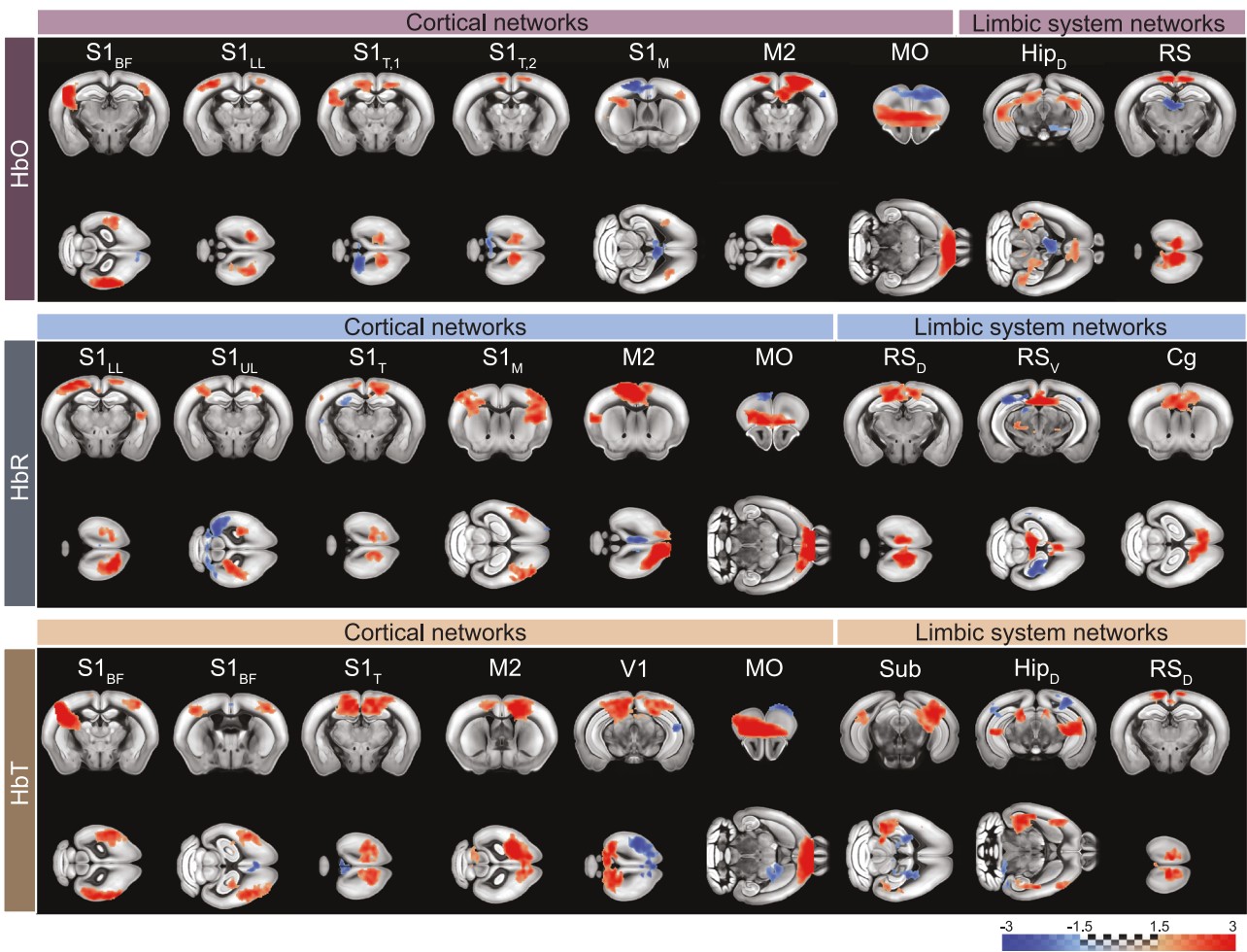

**Fig. 5 | Group independent component analysis (ICA) components identified from the oxygenated (HbO), deoxygenated (HbR), and total (HbT) hemoglobin recordings.** Functional optoacoustic (fOA) components revealed cortical and limbic networks. The fOA ICA maps unveiled unique patterns within each hemodynamic component, showcasing spatially overlapping resting-state networks between the two modalities. $S1_{BF}$ = primary somatosensory area, barrel field; $S1_{LL}$ = primary somatosensory area, lower limb; $S1_{UL}$ = primary somatosensory area, upper limb; $S1_T$ = primary somatosensory area, trunk; $S1_M$ = primary somatosensory area, mouth; M2 = secondary motor area; V1 = primary visual area; MO = orbital area, medial part; Cg = anterior cingulate area; $RS_D$ = retrosplenial area, dorsal part; $RS_V$ = retrosplenial area, ventral part; Sub = Subiculum; $Hip_D$ = dorsal hippocampus.

networks were revealed from BOLD, while ten networks were distinguishable by analyzing the HbO, HbR, and HbT components individually. The hindered sensitivity of fOA to deeper brain regions, compared to BOLD, likely accounts for a large number of non-overlapping networks, with fOA predominantly identifying fine-grained superficial networks while BOLD captures more distributed networks across the entire brain. A comparison of BOLD and hemoglobin-based fOA components unveiled intriguing correlations and subtle discrepancies, particularly in anti-correlations. These findings highlight the complex interplay of different hemodynamic factors in brain connectivity and emphasize the importance of considering the contributions of different hemoglobin states when interpreting functional neuroimaging data. The use of a hybrid MROT system proves essential in this regard, offering critical advantages over standalone systems by enabling the simultaneous acquisition of both imaging modalities. This ensures precise temporal and spatial alignment, crucial for simultaneously capturing dynamic physiological processes such as neurovascular coupling and hemodynamic changes. In addition, concurrent data collection minimizes coregistration discrepancies caused by subject motion, changes in orientation, or tissue deformation. By leveraging a hybrid system, we can further capture complex, transient biological events

in real time, providing a more integrated perspective on multifaceted brain activity. Relying solely on independent systems may risk overlooking such events occurring at fine temporal scales.

Prior research on combined fMRI-EEG has shown a close connection between spontaneous neuronal activity and BOLD fluctuations during rest[52,53]. Building on this, the observation of a strong linear relationship between the voxel-wise functional connectivity correlations derived from BOLD and fOA further strengthens our understanding of the link between neural activity and hemoglobin concentration. The consistency of this pattern, evident across diverse functional brain regions and individuals, underscores oxygenation-dependent fOA components as reliable markers for accurately mapping functional brain connectivity.

Traditionally, the prevailing assumption in the neuroimaging field has been that fluctuations in HbR primarily drive BOLD signal changes. This perspective derives from the BOLD signal nature being primarily attributed to the paramagnetic properties of HbR and its impact on the local magnetic field[42]. Despite the known influence of HbR on magnetic susceptibility, studies have shown that even under high static magnetic fields (up to 18 T), its chromatic absorption properties remain largely unchanged[54]. This indicates that static magnetic field strength has minimal impact on the rsFC mapping of fOA, suggesting the robustness

of these measurements against magnetic interference. Findings from fNIRS-fMRI studies in sensory-evoked responses exhibited variability, with some indicating a higher spatiotemporal correlation of BOLD and HbR[55,56] whilst others showed equal correlations with both HbO and HbR components[57] or a stronger correlation with HbT[43,58,59]. It is important to note that most of these studies have not fully investigated the interactions between all the hemoglobin components (HbO, HbR, HbT)[60], thus hampering a proper comparison. In this context, rsFC becomes a powerful tool for investigating the relationship between hemoglobin concentrations and BOLD signals. Preliminary research in multimodal rsFC points to HbT as a more effective marker for discerning functional connectivity[61–64]. However, limitations in spatial resolution and penetration depth inherent in fNIRS and ISOI hamper an effective analysis of spatial correlations with BOLD signals[65].

By unlocking high spatiotemporal resolution at depths, fOA stands as a promising approach for elucidating the connection between BOLD and hemoglobin responses. The results obtained in this work indicate that HbT aligns with BOLD by capturing both positive and negative connections consistently throughout the acquisition, confirming its potential as a comprehensive marker for assessing functional connectivity. Since hemoglobin is confined to the red blood cells with the vast majority of oxygen in the blood bound to hemoglobin, HbT is considered an indicator reflecting changes in the local blood volume, allowing for the direct characterization of local perfusion and vasomotor activities[66,67]. Thus, these findings may underscore the importance of accurate cerebral blood volume measurements as an indicator of rsFC. In addition, similar correlation strengths between HbO, HbT, and BOLD further emphasize their shared ability to detect functional brain networks. Considering the observed relatively low correlation of HbR with BOLD, the assumed relationship between HbR and BOLD signals may have oversimplified the intricate rsFC and stimulus-evoked mechanisms, potentially overlooking the contributions from other hemodynamic factors. The observed heightened noise in HbR rsFC mapping could further potentially compromise its utility in accurately assessing functional connectivity. This new insight encourages a more integrative approach considering the roles of both HbO and HbR for a holistic understanding of cerebral hemodynamics.

The multiparametric imaging performance of MROT can further be enhanced with upcoming technical developments. Advanced acquisition, reconstruction, and processing methodologies may further refine the identification of fOA rsFC networks. To this end, the limited penetration of light into biological tissues results in fOA achieving shallower penetration depths than fMRI, as evidenced by the distinct absence of subcortical counterparts in the ICs derived from fOA data. Optimizing light delivery into the brain can potentially improve the sensitivity of fOA in deeply embedded areas[68]. Refined image reconstruction methodologies also play an important role in quantifying rsFC with fOA, as evinced by the superior performance of the non-negative constraint model-based image reconstruction methodology[46]. At present, the fOA imaging performance is impeded by multiple other factors, such as per-pulse laser energy and beam shape fluctuations, as well as wavelength-dependent light attenuation, which may need to be properly accounted for in order to achieve a more accurate characterization of functional connectivity. While both fOA and fMRI detect hemodynamic responses, fluorescence imaging of mice expressing calcium or voltage indicators can additionally enable the examination of functional connectivity at the level of individual neurons or specific cell populations, thus providing more direct information on neuronal activity[32,69]. Future efforts may then focus on the simultaneous integration of fluorescence, fOA, and fMRI readings to unravel the intricate interplay among multiple cell types metabolic and vascular responses.

In summary, the findings in this work demonstrate the unique capability of MROT to provide new insights into rsFC in mice. The observed correlations between BOLD and fOA-resolved hemodynamic components underscore the potential of fOA to complement and enrich the understanding of functional brain networks. The observation that HbT and HbO components have a stronger correlation with BOLD signals than HbR supports the view that the BOLD signal is largely due to vasodilation and increased blood volume, thus opening new avenues for reassessing the role of hemoglobin-based contrast in functional neuroimaging. These findings lay the groundwork for future investigations aimed at elucidating the fundamental nature of functional brain connectivity, neurovascular, and neurometabolic coupling mechanisms and mark a significant leap forward in the quest for the identification of specific biomarkers of neurological disorders.

## Methods

### Hybrid magnetic resonance optoacoustic tomography (MROT) system

The MROT system integrates an OA imaging module with a 9.4 T high-field MRI scanner (BioSpec 94/20, Bruker BioSpin, Ettlingen, Germany). OA imaging was performed with an MRI-compatible spherical matrix transducer array (Imasonic SAS, Voray, France) consisting of 384 piezocomposite elements with a central frequency of 5 MHz. These elements are arranged across a hemisphere with a 40 mm radius and 130° angular aperture, resulting in a nearly isotropic resolution of ~163 μm[36]. The array was housed in plastic with polyetheretherketone (PEEK) material to ensure MRI compatibility. OA excitation was achieved with a short-pulsed (<10 ns) optical parametric oscillator (OPO) laser (SpitLight, Innolas Laser GmbH, Germany). The laser was configured to rapidly alternate between several wavelengths at 50 Hz pulse repetition frequency with ~8 mJ per pulse energy. Light delivery to the target tissue was achieved with an MRI-compatible fiber bundle (CeramOptec GmbH, Bonn, Germany) made of fused silica tubes with ferrules made of polyoxymethylene (POM) material and a protection tube made of brass. The fiber bundle was inserted in an 8 mm aperture in the transducer array. Acoustic coupling was facilitated by a PEEK cap with a 36 mm diameter opening sealed with an optically and acoustically transparent polyethylene membrane. Deuterium oxide (heavy water) was used as a coupling medium in the enclosed volume, while ultrasound gel mixed with deuterium oxide was used to ensure acoustic coupling to the membrane. Data acquisition was managed with a custom-built DAQ system (Falkenstein Mikrosysteme GmbH, Germany) capable of sampling at 40 mega samples per second (MSPS). A pair of custom radiofrequency (RF) coils, integrated into the MRI setup, were positioned on either side of a 3D-printed animal holder. These coils were used to generate a magnetic field orthogonal to the main magnetic field of the scanner. An external trigger device (Pulse Pal V2, Sanworks, USA) was used to synchronize the Q-switch output of the laser with the beginning of the MRI acquisition for precise temporal correlation.

### MRI data acquisition

MRI scanning was performed with a 9.4 T Bruker Biospec 94/20 small animal MR system using custom-made RF coils as described above. $T_1$-weighted scans along the coronal plane serving as the anatomical reference were obtained using a fast low-angle-shot (FLASH) sequence: field of view (FOV) = 20 × 10 mm², matrix dimension (MD) = 160 × 80, 11 slices, slice thickness = 0.7 mm, repetition time (TR) = 500 ms, echo time (TE) = 2.1366 ms, and number of averages (NA) = 8. MR angiography (MRA) images were further acquired to facilitate image coregistration with OA using a 2D- time-of-flight (TOF) sequence: FOV = 20 × 20 mm², 20 slices, slice thickness = 0.3 mm, TR = 13 ms, TE = 1.8904 ms, flip angle (FA) = 80°, NA = 16. Prior to fMRI data acquisition, the local field homogeneity was optimized using the acquired $B_0$ field maps. For the rsFC acquisitions, volumes were acquired using a gradient-echo echo-planar imaging (GE-EPI) sequence: FOV = 20 × 10 mm², MD = 80 × 40, 11 slices, slice thickness = 0.7 mm, TR = 995 ms, TE = 12 ms, FA = 60°, temporal resolution = 1 s. Scan durations varied between 600 s–980 s per dataset.

## fOA data acquisition

fOA data acquisition was performed simultaneously with the fMRI data acquisition with identical scan durations. Volumetric time-lapse fOA data were acquired at 700, 730, 755, 800, and 850 nm excitation wavelengths. Following the acquisition, sinogram restoration was performed to address RF-induced interference[37]. The raw signals were subsequently bandpass filtered between 0.1 and 8 MHz. Linear spectral unmixing enabled the extraction of time-resolved HbO and HbR distributions in the brain, with HbT calculated as the sum of unmixed HbO and HbR components. Image reconstruction was performed with an iterative model-based algorithm featuring a non-negativity constraint to avoid negative image values with no physical meaning[45,46,70]. A voxel size of $100 \times 100 \times 100 \ \mu m^3$ and a FOV of $8 \times 8 \times 4 \ mm^3$ at the center of the spherical array geometry were considered for image reconstruction. Images were normalized to the laser pulse energy readings at the corresponding wavelengths and further normalized with an exponential light attenuation function to compensate for signal intensity decay with depth[71].

## In vivo imaging

Athymic female nude mice (Foxn1[nu], Charles River Laboratories, Germany, 9-18-week-old, $n = 16$, female) were imaged in this study with the in vivo data successfully recorded for all the animals. The animals were housed in individually ventilated, temperature-controlled cages under a 12-h reversed dark/light cycle. Pelleted food (3437PXL15, CARGILL) and water were provided *ad libitum*. Anesthesia was induced with intraperitoneal (i.p.) injection of a mixture of ketamine (100 mg kg$^{-1}$ body weight, Pfizer) and xylazine (10 mg kg$^{-1}$ body weight, Bayer). Maintenance injection was administered i.p. every 45 min, consisting of a mixture of ketamine (25 mg kg$^{-1}$ body weight) and xylazine (1.25 mg kg$^{-1}$ body weight). Both the scalp and the skull of the mice were kept intact during the experiments. Mice were positioned onto the 3D-printed mouse bed in a prone position with the nose pointing downwards. During the experiment, an oxygen/air mixture (0.2/0.8 L min$^{-1}$) was provided through a breathing mask. Body temperature and respiration were continuously monitored during data acquisition with an MRI-compatible rectal thermometer and a pneumatic pillow (SA Instruments, USA). The heart rate and SpO$_2$ were monitored in real-time with an MRI-compatible mouse paw pulse oximeter (PhysioSuite, Kent Scientific Corporation, USA). The body temperature was kept around 37 °C with a temperature-controlled water heating unit. Mouse housing, handling, and experimentation were performed in accordance with the Swiss Federal Act on Animal Protection and were approved by the Cantonal Veterinary Office Zurich.

## Data preprocessing

BOLD, HbO, HbR, and HbT datasets were pre-processed separately (Supplementary Fig. 21). rs-fMRI and rs-fOA datasets were corrected for motion and smoothened with a Gaussian kernel using SPM12 (Wellcome Trust Center for Neuroimaging, London, UK). Mean framewise displacements (FWDs) calculated from BOLD and HbO images ranged between 9.8–45.4 $\mu m$ and between 12.2–40 $\mu m$, respectively (Supplementary Fig. 22). The anatomical and functional images underwent co-registration and spatial normalization to the Allen Brain Institute reference atlas, as previously described[38,47]. Each preprocessing output was visually inspected before further analysis. The data of one mouse were excluded from further analysis due to susceptibility artifacts in the fMRI image. fMRI datasets underwent bandpass filtering (0.01–0.1 Hz), detrending, and motion parameters + ventricle + vascular regression, which was shown to enhance the functional connectivity specificity[49]. The reconstructed time-lapse fOA datasets were band-pass filtered to a narrower 0.03–0.1 Hz frequency band to suppress the spurious connectivity due to the presence of laser fluctuation signals peaking at ∼ 0.02 Hz (Supplementary Fig. 23). The vascular signal and the motion parameters were regressed out.

The assessment of resting-state dataset quality involved scrutinizing the simultaneous presence of robust inter-hemispheric connectivity in sensory cortices (left S1$_{BF}$ to right S1$_{BF}$, r$_{FC}$ > 0.1), coupled with either weak connectivity or anti-correlation between sensory areas and anterior cingulate areas (left S1$_{BF}$ to ACA, r$_{FC}$ < 0.1)[48]. The datasets meeting this criterion (10 out of 15 datasets) were included in further analysis (Supplementary Fig. 2).

## Resting-state functional connectivity analysis

Functional connectivity was first assessed using cortical seed regions measuring $0.4 \times 0.4 \times 0.4 \ mm^3$ each, as defined by the Allen Common Coordinate Framework (CCF). Seed-to-voxel functional connectivity maps were computed for each animal. These maps were based on pre-processed time series data extracted for BOLD, HbO, HbR, and HbT signals, with connectivity quantified using Pearson's correlation coefficient. At the group level ($n = 10$), average seed connectivity maps were generated by combining data from individual animals. The spatial coherence and smoothness of the connectivity maps were assessed using local variance and average spatial gradients. The local variance was computed using a sliding window ($3 \times 3 \times 3$ voxels) around each voxel, capturing the variability of rsFC values within that neighborhood. Higher local variance indicated noisier and less coherent connectivity patterns. Gradient-based spatial consistency was quantified by averaging the magnitudes of spatial gradients in the x, y, and z-directions. Higher gradient magnitudes pointed to larger fluctuations and, consequently, less smooth spatial patterns in the seed connectivity maps.

Region-to-region connectivity matrices were constructed for each component (i.e., BOLD, HbO, HbR, and HbT) using an extended set of regions of interest from the Allen CCF. Voxel-to-voxel connectivity matrices were separately generated for each hemodynamic component, focusing specifically on gray matter voxels. The origin of each voxel was mapped according to the Allen CCF to ensure anatomical consistency. Group-level independent component analysis (ICA) was carried out separately for each hemodynamic component using the CONN toolbox[72] with 20 components.

## Correlation between modalities and statistical analysis

Normalized cross-correlations (NCCs)[38] between connectivity maps for BOLD and fOA components (HbO, HbR, HbT) were calculated to evaluate spatial correlation. We segregated the connectivity maps into two categories: those with positive correlations, and those with only negative correlations relative to the seed signal. Within the fOA components, paired t tests were employed to identify significant differences in similarity to the BOLD maps, both at individual and group levels. Effect sizes for the group level differences were quantified using Cohen's d with significance of $\alpha = 0.05$. The connectivity pairs from BOLD and fOA components were plotted at the voxel level, and the correlation coefficient of the variables was calculated for HbO, HbR, and HbT for each seed, both individually and at the group level. This method was similarly applied to the region-to-region connectivity edges.

## Reporting summary

Further information on research design is available in the Nature Portfolio Reporting Summary linked to this article.

# Data availability

All data supporting the findings of this study are found within the paper and its Supplementary Information. Source data are provided in this paper.

# Code availability

The code that supports the findings of this study is available from the corresponding author upon request.

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

## Acknowledgements

The authors acknowledge support from the Swiss National Science Foundation grant 310030_192757 (D.R.). V.Z. acknowledges funding support by the Swiss National Science Foundation (SNSF) ECCELLENZA (PCEFP3_203005).

## Author contributions

D.R., I.G., and Z.C. conceived the concept. I.G. and Z.C. performed the animal experiments with the help of H.A.I.Y., and X.L.D. I.G. performed data analysis and visualizations. V.Z., Z.C., X.L.D., and H.A.I.Y. provided guidance on experimental procedures and data analysis. D.R. supervised the work. All authors contributed to writing and revising the manuscript.

## Competing interests

The authors declare no competing interests.
