## [Transparent Peer Review file · Nature Communications]

Concurrent optoacoustic tomography and magnetic resonance imaging of resting-state functional connectivity in the mouse brain

Corresponding Author: Professor Daniel Razansky

Version 0:

Reviewer comments:

Reviewer #1

(Remarks to the Author)

This is a very interesting study on resting-state functional connectivity (rsFC) acquired by optoacoustic tomography (OAT) and MRI BOLD simultaneously in a lightly anesthetized mouse model. The authors performed seed-based and independent component analyses, revealing strong correlations between MRI BOLD signals and all types of hemoglobin, suggesting that OAT may be a valuable tool for rsFC in rodents. I have a few comments and questions.

The fMRI data acquisition took 600-980 seconds. How long was the OAT acquisition? Assuming it was much shorter, was it acquired at the beginning, middle, or end of the fMRI acquisition? Did the authors try analyzing the OAT data at different times, and did they notice any difference in OAT signals between the beginning and end of the fMRI acquisition? It's unclear if the fMRI results have accounted for possible hysteresis, but I suspect OAT might shed light on this.

MRI involves the production of large magnetic fields. Do these fields affect the chromatic absorption of hemoglobin, particularly deoxygenated hemoglobin (HbR), in a meaningful way? Have the authors performed a control to determine whether HbR, oxyhemoglobin (HbO), and total hemoglobin (HbT) mapping looks the same in the presence and absence of fMRI acquisition?

Other research groups have found higher quality connectivity with less noise in HbO and HbT rsFC mapping compared to HbR (e.g., source using intrinsic optical signal imaging). Do their results parallel those found in your paper?

Independent component analysis (ICA) identified 20 resting-state networks in the BOLD system, including 12 in the cortical and limbic system networks. HbO identified 9 networks, with only 3 overlapping with BOLD; HbR identified 9 networks, with 2 overlapping with BOLD; and HbT found 9 networks, with 5 overlapping with BOLD. How do you account for the large number of networks identified by BOLD and OAT that are non-overlapping, particularly within the regions of the brain presumably visualizable by OAT?

You have previously demonstrated the ability to differentiate HbO and HbR using OAT in mice. Have you fully considered the effect of wavelength-dependent light attenuation on the unmixing algorithms? Do you see better HbR correlation with BOLD in regions with less light attenuation?

The rationale for BOLD as a measure of HbR is that deoxygenated hemoglobin is an indirect measure of neural activity as it reflects oxygen extraction and consumption by neurons. Does your manuscript argue that BOLD is actually reporting HbT or HbO, not HbR? If so, how do your results "underscore oxygenation-dependent fOA components as reliable markers"?

Reviewer #2

(Remarks to the Author)

The manuscript effectively highlights the limitations of current fMRI-based studies in understanding resting-state functional connectivity (rsFC) due to the complex physiological origins of the BOLD signal. The development of hybrid magnetic resonance optoacoustic tomography (MROT) was achieved by implanting an MRI-compatible spherical matrix transducer array and MRI-compatible optical fiber. Additionally, an optical and ultrasound transparent membrane was used for coupling in animal brain experiments. The use of MROT for concurrent multiparametric hemodynamic recordings is innovative. The results are meticulously detailed, showcasing the strengths of fOA imaging in revealing spatially overlapping bilateral correlations between fMRI-BOLD and hemodynamic components. The observation of stronger correlations between BOLD and total hemoglobin (HbT) and oxygenated hemoglobin (HbO) than with deoxygenated hemoglobin (HbR) is particularly noteworthy. This finding challenges conventional assumptions and underscores the importance of considering multiple hemodynamic factors in neuroimaging studies. Overall, this manuscript represents a significant contribution to the field of neuroimaging and functional connectivity studies. It successfully demonstrates the potential of hybrid imaging approaches to provide deeper insights into brain function and connectivity. However, there are minor weaknesses that can be improved after revision:

1. Please highlight why developing a hybrid magnetic resonance optoacoustic tomography (MROT) system is necessary when image registration could be done between independent MRI and optoacoustic imaging systems with the help of AI algorithms. Combining both MRI and optoacoustic imaging in one system would be very expensive, and MRI-compatible components are needed.
2. Since both MRI and optoacoustic imaging provide similar information on blood flow followed by neuron activities and have real-time imaging capabilities with optoacoustic imaging, why is the dynamic information not shown in this manuscript?
3. The optoacoustic imaging in Supplementary Figure 3 provides relatively low-resolution images, making them hard to interpret. More explanation is needed for the different color maps.
4. The voxel-to-voxel connectivity matrices need more details and citations.
5. More details about the MRI-compatible spherical matrix transducer array and MRI-compatible optical fiber are needed. What materials are they made of?

Version 1:

Reviewer comments:

Reviewer #1

(Remarks to the Author)

The authors have responded to all my comments.

Reviewer #2

(Remarks to the Author)

Thank you to the authors for thoroughly addressing all my questions. Excellent work!

Point-by-point responses to the Reviewers' comments

Reviewer #1:

This is a very interesting study on resting-state functional connectivity (rsFC) acquired by optoacoustic tomography (OAT) and MRI BOLD simultaneously in a lightly anesthetized mouse model. The authors performed seed-based and independent component analyses, revealing strong correlations between MRI BOLD signals and all types of hemoglobin, suggesting that OAT may be a valuable tool for rsFC in rodents. I have a few comments and questions.

Reply: We appreciate the reviewer's positive feedback and encouraging comments. Please find the point-by-point responses to the comments below.

The fMRI data acquisition took 600-980 seconds. How long was the OAT acquisition? Assuming it was much shorter, was it acquired at the beginning, middle, or end of the fMRI acquisition? Did the authors try analyzing the OAT data at different times, and did they notice any difference in OAT signals between the beginning and end of the fMRI acquisition? It's unclear if the fMRI results have accounted for possible hysteresis, but I suspect OAT might shed light on this.

Reply: We thank the reviewer for highlighting this point. OAT and fMRI data acquisition were synchronized and performed simultaneously, i.e., OAT data acquisition also took 600-980 seconds. We now updated the corresponding sections to clarify this point further – see in lines 94 and 365 in the revised manuscript.

Regarding the analysis of OAT data at different times during the fMRI acquisition and the potential impact of hysteresis, we have revisited the OAT data with a focus on temporal variation. For this, a detailed analysis was performed by dividing the data into the first and second halves, examining patterns in both stand-alone OAT signals and BOLD-fOA correlations across these segments. This analysis has been included in the new supplementary figures as follows: rsFC maps for each temporal segment and correlation across hemodynamic channels for the two halves of the time-lapse data (new SFig. 4), voxel-wise correlations between fOA components and BOLD (new SFig. 9), and cross-correlations between fOA components and BOLD (new SFig. 15).

Our findings indicate that the spatial overlap of rsFC patterns persisted across temporal segments (new SFig. 4). Notably, correlations between rsFC maps derived from the first and second halves of the time interval of individual hemodynamic components were higher for HbO and HbT compared to HbR and BOLD, likely due to the inherently noisier nature of HbR and BOLD signals (new SFig. 4). In addition to the lower correlation between the two halves of the time-lapse data, we also observed that the seed maps generated from HbR data were consistently noisier compared to those from HbO and HbT (new SFig. 7).

We also observed that voxel-wise correlations between fOA components and BOLD were lower when considering only half of the acquisition duration (new SFig. 9). This is expected due to the increased noise and variability in shorter time windows. This highlights the importance of sufficient acquisition duration to achieve robust correlations between BOLD and fOA components. Further cross-correlation analyses of seed maps from the first and second halves of the data, similar to those in Fig. 2g-h, consistently showed stronger correlations between HbO/HbT and BOLD compared to HbR across both positive and negative connections (new SFig. 15). This consistency suggests that while some temporal variability exists, particularly in noisier signals like HbR and BOLD, the overall rsFC patterns remain stable.

Although we did not explicitly model hysteresis in our initial analysis, the consistency of findings across the first and second halves suggests that any hysteresis effects, if present, do not significantly impact our primary conclusions. The observed stability in HbO and HbT correlations with BOLD further supports the robustness of these hemodynamic measures, even when data is temporally segmented.

We have updated our manuscript to incorporate these findings. Please refer to the new SFig. 4, SFig. 7, SFig. 9, SFig. 15, and the revised text in lines 115, 150, 178, 231, 279, 289.

MRI involves the production of large magnetic fields. Do these fields affect the chromatic absorption of hemoglobin, particularly deoxygenated hemoglobin (HbR), in a meaningful way? Have the authors performed a control to determine whether HbR, oxyhemoglobin (HbO), and total hemoglobin (HbT) mapping looks the same in the presence and absence of fMRI acquisition?

Reply: We appreciate the Reviewer's question. A previous study investigating deoxygenated hemoglobin under high static magnetic fields (up to 18 Tesla) revealed that the magnetic field caused a slight (~1%) enhancement in absorption peaks without shifting them or altering the overall spectral trend [1]. This indicates that while the magnetic field has a minor influence on absorption intensity, it does not fundamentally change the chromatic absorption properties of hemoglobin. Hence, any magnetic field effects on absorption spectra would be negligible, especially in comparison to the inherent differences between the absorption spectra of HbO and HbR across different wavelengths. As such, this would not lead to any variations in rsFC mapping. We updated our discussion in line 264 to further clarify this point.

We have additionally conducted control experiments to assess to which extent our results are impacted by the magnetic gradients involved in fMRI acquisitions. The subsequent analysis confirmed that the seed-maps from a single subject remained highly consistent across all fOA components, whether acquired concurrently with fMRI or in standalone fOA sessions (new SFig. 5). It is important to note that the time difference between the start of the concurrent and standalone acquisitions was approximately 30 minutes. This temporal gap has introduced inevitable variability in rsFC mappings due to factors such as the anesthesia regime, biological fluctuations, and the physiological status of the mouse. The correlations of the constructed seed-maps across the two acquisition modes showed no significant differences ($p = 0.11$ for HbO~HbR, $p = 0.15$ for HbT~HbR). However, the correlation of HbR across acquisitions was found lower, albeit not statistically significant (n.s.), compared to HbO and HbT. This finding further highlights the heightened susceptibility of HbR signals to noise, as observed in both the noise quantification analysis (new SFig. 7) and the between-halves stability of the rsFC maps (new SFig. 4).

We have updated the corresponding parts of the manuscript (line 120) and presented the extended results in new SFig. 5.

[1] M Mauricy et al., *Bioelectrochemistry and Bioenergetics* 47 (2), 297 (1998).

Other research groups have found higher quality connectivity with less noise in HbO and HbT rsFC mapping compared to HbR (e.g., source using intrinsic optical signal imaging). Do their results parallel those found in your paper?

Reply: Indeed, we have found noisier rsFC mapping in HbR, compared to HbO and HbT, thus paralleling previous reports that employed intrinsic optical signal imaging [2]. In the revised manuscript, we quantified the smoothness of noise present in the rsFC patterns by analyzing spatial properties of the correlation maps. Specifically, we estimated the local variance and gradient-based spatial consistency for the 14 seed-maps at the group level ($n=10$) to assess the noise levels. The results indicated higher local variance and larger fluctuations in the HbR connectivity maps, compared to HbO and HbT. We have updated the corresponding parts of the manuscript (lines 125 and 426) and displayed the results in the new SFig. 7.

[2] S Kura et al., *Journal of neural engineering* 15 (3), 035003 (2018).

Independent component analysis (ICA) identified 20 resting-state networks in the BOLD system, including 12 in the cortical and limbic system networks. HbO identified 9 networks, with only 3 overlapping with BOLD; HbR identified 9 networks, with 2 overlapping with BOLD; and HbT found 9 networks, with 5

overlapping with BOLD. How do you account for the large number of networks identified by BOLD and OAT that are non-overlapping, particularly within the regions of the brain presumably visualizable by OAT?

Reply: We thank the Reviewer for raising this important point. The large number of non-overlapping networks identified by BOLD fMRI and OAT can primarily be attributed to the inherent differences in depth sensitivity and signal characteristics between the two modalities. OAT is constrained by its limited penetration, as the strong light absorption and scattering significantly diminish the signal strength with depth. Consequently, this limits the effective visualization of deeper brain structures, making OAT less suitable for detecting networks originating from deeper areas of the brain. In contrast, BOLD fMRI can capture hemodynamic changes across the entire brain, including those deeper areas.

Due to OAT's better sensitivity in the shallow brain regions, the networks identified using HbO, HbR, and HbT tend to be fine-grained and localized to superficial layers, such as the smaller regions within the somatosensory cortex, when using ICA with 20 components. These networks predominantly arise from the cortical areas rather than from deeper regions within the field of view. BOLD fMRI, on the other hand, captures signals from more diverse regions across the brain, resulting in coarser and more distributed networks when using the same 20-component ICA decomposition. While it is possible to extract finer-grained networks from BOLD fMRI by increasing the number of ICA components, this approach often leads to the detection of a greater number of unilateral components, which may complicate interpretation of the results.

In the revised manuscript, we added a discussion in line 237 to highlight this issue.

You have previously demonstrated the ability to differentiate HbO and HbR using OAT in mice. Have you fully considered the effect of wavelength-dependent light attenuation on the unmixing algorithms? Do you see better HbR correlation with BOLD in regions with less light attenuation?

Reply: We appreciate this suggestion. Indeed, achieving accurate light fluence attenuation correction in unmixing algorithms is a long-standing goal in the biomedical optics field and in optoacoustic imaging in particular. This is known to be particularly challenging in the context of *in vivo* imaging for several reasons. For example, slight variations in the position of the mouse during the imaging session relative to the incident light beam (approximately Gaussian) can lead to significant changes in the light distribution both at the surface and at deep regions. More importantly, intrinsic heterogeneities of biological tissues are too complex to be accurately modeled and can significantly vary between animals. The multilayered structure of the mouse head, including scalp, skull, and brain, poses additional challenges to achieve an accurate estimation of the light fluence distribution. Note also that the optical properties of biological tissues available in the literature have been measured *ex vivo*, and can significantly change in living animals.

Considering the aforementioned challenges, we strived to account for the most prominent sources of uncertainty and carefully analyzed the effects of this correction in the results obtained. Specifically, the optoacoustic signals were normalized by the light energy density at each wavelength, estimated with an India ink absorber located approximately at the position of the mouse skin surface. This approach helps to account for variations in light delivery across the spectrum, providing a baseline correction for wavelength-dependent attenuation. In addition to this, we have conducted additional analysis in our revised manuscript by applying an exponential correction model based on depth, considering the expected (average) wavelength dependence of absorption and scattering in biological tissues. This method specifically addresses the increased attenuation at greater depths, where light intensity is known to decrease exponentially for a uniform light distribution at the surface. By incorporating this additional correction, we aimed to assess how the rsFC patterns would be affected.

Our new results demonstrated a strong correlation of rsFC patterns between the energy-normalized data and the data further corrected with the depth-dependent attenuation model (new SFig. 16). The seed-maps

remained largely consistent, and the correlation across these maps was high ($r > 0.96$ for all fOA components). Although the differences were statistically insignificant, we observed a slightly lower correlation for HbO and HbR compared to HbT (new SFig. 16).

To comprehensively understand the impact of these fluence correction strategies on the overall analysis, we further performed voxel-based correlations and cross-correlation with BOLD signals. Interestingly, while the depth-dependent model resulted in a minor increment in voxel-based correlations, this effect was observed across all components and was ultimately negligible (new SFig. 17). Additionally, the differences in cross-correlation with BOLD signals were found to be largely independent on the fact that exponential correction was performed, indicating robustness in our findings (new SFig. 18).

We acknowledge the importance of continued research to improve fluence correction techniques, particularly in the context of the hybrid MROT imaging method. The concurrent imaging capabilities of MROT offer a unique opportunity to model the complex nature of the brain more accurately by minimizing the effects of variables such as mouse positioning. This may be achieved by leveraging the anatomical information provided by MRI to develop a more sophisticated model for estimating fluence, although as mentioned above, the estimation of optical properties in segmented regions remains a challenge.

To further assess the effect of depth on the correlation with BOLD signals, we conducted a separate analysis that examined the correlation of HbO, HbR, and HbT at different depths. The difference between HbT and HbR correlations was more pronounced in regions with less light attenuation that are closer to the surface of the brain (new SFig. 11). This trend was also evident when comparing HbO and HbR correlations, and it was consistent across both positive and negative connections. These findings reinforce the conclusions presented in the manuscript.

We have updated our manuscript to incorporate these findings. Please refer to the new SFig. 11, SFig. 16, SFig. 17, SFig. 18 and the revised text in lines 172 and 178.

The rationale for BOLD as a measure of HbR is that deoxygenated hemoglobin is an indirect measure of neural activity as it reflects oxygen extraction and consumption by neurons. Does your manuscript argue that BOLD is actually reporting HbT or HbO, not HbR? If so, how do your results “underscore oxygenation-dependent fOA components as reliable markers”?

Reply: We apologize for any confusion regarding the claims made. Our manuscript does not argue that BOLD primarily reflects HbT or HbO instead of HbR. Rather, we recognize that the BOLD signal is inherently multifaceted, being shaped by multiple hemodynamic factors, including HbR, HbO, and HbT. While BOLD has traditionally been associated with changes in HbR due to its paramagnetic properties and its influence on the local magnetic field, our findings indicate that HbO and consequently HbT also play important roles in modulating the BOLD signal, particularly in the context of resting-state functional connectivity dynamics.

Our results suggest that all the oxygenation-dependent fOA components provide valuable complementary information, capturing different aspects of the hemodynamic response to neuronal activation. In this context, we emphasize that while BOLD remains closely linked to HbR, the inclusion of HbO and HbT measurements through fOA enhances our understanding of the broader oxygenation dynamics at play and reinforces the reliability of fOA components as markers of neural activity. Notably, we observed a stronger link between HbO and HbT to BOLD while capturing the resting-state functional connectivity measures, suggesting that functional connectivity at rest may be more strongly influenced by cerebral blood volume and oxygenation dynamics rather than HbR alone. Thus, rather than diminishing the role of HbR, our findings support a more integrative approach that incorporates the contributions of all the hemodynamic components alongside HbR to more comprehensively elucidate the mechanisms underlying the BOLD signal and its relationship to brain activity.

Reviewer #2:

The manuscript effectively highlights the limitations of current fMRI-based studies in understanding resting-state functional connectivity (rsFC) due to the complex physiological origins of the BOLD signal. The development of hybrid magnetic resonance optoacoustic tomography (MROT) was achieved by implanting an MRI-compatible spherical matrix transducer array and MRI-compatible optical fiber. Additionally, an optical and ultrasound transparent membrane was used for coupling in animal brain experiments. The use of MROT for concurrent multiparametric hemodynamic recordings is innovative. The results are meticulously detailed, showcasing the strengths of fOA imaging in revealing spatially overlapping bilateral correlations between fMRI-BOLD and hemodynamic components. The observation of stronger correlations between BOLD and total hemoglobin (HbT) and oxygenated hemoglobin (HbO) than with deoxygenated hemoglobin (HbR) is particularly noteworthy. This finding challenges conventional assumptions and underscores the importance of considering multiple hemodynamic factors in neuroimaging studies. Overall, this manuscript represents a significant contribution to the field of neuroimaging and functional connectivity studies. It successfully demonstrates the potential of hybrid imaging approaches to provide deeper insights into brain function and connectivity. However, there are minor weaknesses that can be improved after revision:

Reply: We appreciate the Reviewer's positive feedback and encouraging comments. Please find the point-by-point responses to the comments below.

1. Please highlight why developing a hybrid magnetic resonance optoacoustic tomography (MROT) system is necessary when image registration could be done between independent MRI and optoacoustic imaging systems with the help of AI algorithms. Combining both MRI and optoacoustic imaging in one system would be very expensive, and MRI-compatible components are needed.

Reply: We thank the Reviewer for this insightful question. Developing a hybrid MROT system is essential despite the availability of algorithms for image registration between independent MRI and optoacoustic imaging systems. While standalone systems combined with AI-based registration might seem like a cost-effective alternative, there are several key advantages to having a truly hybridized MROT system. First, MROT allows for the simultaneous acquisition of MRI and optoacoustic data, ensuring precise temporal and spatial alignment of the datasets without the need to reposition the animal and perform a separate measurement. Real-time integration of multimodal neuroimaging data is particularly crucial for capturing dynamic physiological processes, such as neurovascular coupling and hemodynamic changes, where precise temporal synchronization is essential to accurately reflect physiological events. It further eliminates challenges caused by differences in subject motion and biological variability between scans. This is not possible when using standalone systems and post-processing techniques, even with AI assistance.

Second, a hybrid system avoids registration artifacts and alignment errors inherent in separately acquired datasets. Independent acquisition poses significant challenges due to the differences in the relative positions, orientation, and surface deformations of the subjects during separate scans. These effects are particularly pronounced in small animal studies where even slight misalignments may lead to inaccurate registrations. A hybrid MROT system minimizes these issues by acquiring bi-modal images concurrently, eliminating the need for post-hoc registration corrections and reducing artifacts that would otherwise be introduced due to differences in subject positioning and deformation between modalities.

Third, we believe that our hybrid system offers distinct advantages that go beyond what can be achieved through technical realignment alone. What we are trying to address is the complex and multifaceted brain activity. Neural processes, particularly during resting-state conditions, where no clear external stimuli are present, can exhibit temporal variations due to the intricate dynamics of neuronal and glial signaling. These processes are further affected by the presence of various neuromodulators, which can dynamically alter

neurovascular coupling. Relying on independent systems alone may risk overlooking critical transient biological events that occur at fine temporal scales. While standalone systems may suffice for certain research questions, a truly integrated system offers new opportunities to capture and investigate these dynamic processes in real-time, providing a more comprehensive and integrated perspective on the brain function.

As suggested, we have added a discussion on the advantages of performing concurrent fOA and fMRI measurements (line 244).

2. Since both MRI and optoacoustic imaging provide similar information on blood flow followed by neuron activities and have real-time imaging capabilities with optoacoustic imaging, why is the dynamic information not shown in this manuscript?

Reply: In order to address the Reviewer's comment and better demonstrate the dynamic imaging capabilities of the MROT system, we have added new SFig. 3 where we have plotted exemplary time-resolved fOA and BOLD signals from the visual and motor brain areas. These plots illustrate the dynamic nature of the signals, showing how the fOA components (HbO, HbR, HbT) and BOLD signals evolve over time. By displaying signals from both the left and right hemispheres, we also highlight the strong bilateral correlations within each hemodynamic channel. Moreover, the dynamic information presented emphasizes the real-time imaging capabilities of our system. For example, the anti-correlations observed between HbO and HbR within these regions underscore the intricate hemodynamic interactions that are captured by the fOA components.

Additionally, to further utilize the dynamic information provided by the MROT system, we have extended our analysis to investigate how rsFC evolves across different time segments. Specifically, we divided the data into the first and second halves and analyzed the temporal stability of rsFC patterns within these periods (new SFig. 4). Our findings revealed that the rsFC patterns derived from fOA data were generally stable over time, though we observed that correlation between the two halves of the time-lapse data was notably lower for HbR as compared to HbO and HbT. This suggests that HbR, similar to BOLD, might be more susceptible to noise, which can affect the stability of rsFC mapping (please refer to the new SFig. 7 for quantification of noise across fOA components).

To gain further insights into the correlations between fOA components and BOLD at different time points, we conducted voxel-wise correlation analyses and cross-correlations with BOLD maps during the first and second halves of the acquisition (new SFig. 9 and 15). These analyses showed that voxel-wise correlations were lower when the data was divided into shorter time segments, likely due to the increased noise and variability inherent in shorter acquisition windows. Additionally, the cross-correlation analysis consistently demonstrated that HbO and HbT had stronger correlations with BOLD compared to HbR, regardless of the time segment analyzed. This suggests that, while some temporal variability exists, the overall rsFC patterns remain stable and reliable.

We have updated our manuscript to reflect these findings and their implications. Please refer to the new SFig. 3, SFig. 4, SFig. 9, SFig. 15, and the revised text in lines 115, 150, 178, 231, 279, 289.

3. The optoacoustic imaging in Supplementary Figure 3 provides relatively low-resolution images, making them hard to interpret. More explanation is needed for the different color maps.

Reply: We thank the reviewer for highlighting this issue. We have modified SFig. 3 (now SFig. 6 in the revised manuscript) to ensure that the images are of higher resolution, making them easier to interpret. Furthermore, we have standardized the thresholding process to be consistent with the other figures in the manuscript, ensuring a more accurate comparison. We also adapted the limits of the color scale to better emphasize connectivity differences estimated with back-projection reconstruction. Indeed, larger regions

with greater correlation values were observed when using back-projection-based image reconstruction (new SFig. 6) than when using model-based reconstruction (Fig. 2). This is mainly ascribed to the fact that back-projection is prone to generating negative values and other artifacts in the reconstructed images. Streak-type artefacts are particularly common in optoacoustic images reconstructed with back-projection algorithms, often resulting in high intensity arc-shaped regions overshadowing the contrast in surrounding areas. These artifacts can distort the spatial distribution of the signals, causing an overestimation of areas with higher correlation values. Also, back-projection images are more affected by noise, which can further distort certain features in the images. For these reasons, the non-specific elevated intra-hemispheric correlations may arise that do not accurately reflect the underlying rsFC patterns.

We now clarified those points in the text accompanying the new SFig. 6.

4. The voxel-to-voxel connectivity matrices need more details and citations.

Reply: To address the Reviewer's request, we provide the new SFig. 19 for a more detailed presentation of the voxel-to-voxel connectivity matrices. Specifically, we have included a comprehensive color-coded representation of the brain regions from which the analyzed voxels originate. Additionally, we have clarified the methodology in line 437 in the manuscript.

5. More details about the MRI-compatible spherical matrix transducer array and MRI-compatible optical fiber are needed. What materials are they made of?

Reply: In order to make the spherical matrix transducer array compatible with MRI, the housing was fabricated from PEEK plastic instead of metal. Additionally, copper multi-coaxial cables with overall shielding (85Ω) were used. The shielding of the cables, connector shells, and transducer housing was all connected and kept isolated from the electrical ground. The MRI-compatible optical fiber bundle was comprised of a fused silica tube to guide the light, ferrules made of POM plastic, and a protection tube made of brass. As suggested, these details were added in lines 329 and 335 in the revised manuscript.